# Can Event-Related Potentials Evoked by Heel Lance Assess Pain Processing in Neonates? A Systematic Review

**DOI:** 10.3390/children8020058

**Published:** 2021-01-20

**Authors:** Yui Shiroshita, Hikari Kirimoto, Mio Ozawa, Tatsunori Watanabe, Hiroko Uematsu, Keisuke Yunoki, Ikuko Sobue

**Affiliations:** 1Division of Nursing Sciences, Graduate School of Biomedical and Health Sciences, Hiroshima University, Hiroshima 734-8553, Japan; ozawamio@hiroshima-u.ac.jp (M.O.); sobue@hiroshima-u.ac.jp (I.S.); 2Department of Sensorimotor Neuroscience, Graduate School of Biomedical and Health Sciences, Hiroshima University, Hiroshima 734-8553, Japan; hkirimoto@hiroshima-u.ac.jp (H.K.); twatan@hiroshima-u.ac.jp (T.W.); d205546@hiroshima-u.ac.jp (K.Y.); 3School of Nursing, University of Human Environments, Aichi 474-0035, Japan; hirokouematsu0@gmail.com

**Keywords:** heel lance, infant, event-related potential, pain assessment tool

## Abstract

To clarify the possibility of event-related potential (ERP) evoked by heel lance in neonates as an index of pain assessment, knowledge acquired by and problems of the methods used in studies on ERP evoked by heel lance in neonates were systematically reviewed, including knowledge about Aδ and C fibers responding to noxious stimuli and Aβ fibers responding to non-noxious stimuli. Of the 863 reports searched, 19 were selected for the final analysis. The following points were identified as problems for ERP evoked by heel lance in neonates to serve as a pain assessment index: (1) It is possible that the ERP evoked by heel lance reflected the activation of Aβ fibers responding to non-noxious stimuli and not the activation of Aδ or C fibers responding to noxious stimulation; (2) Sample size calculation was presented in few studies, and the number of stimulation trials to obtain an averaged ERP was small. Accordingly, to establish ERP evoked by heel lance as a pain assessment in neonates, it is necessary to perform a study to clarify ERP evoked by Aδ- and C-fiber stimulations accompanied by heel lance in neonates.

## 1. Introduction

Neonates in neonatal intensive care unit (NICU) are exposed to repetitive painful procedures in the period of complex and rapid brain development [1]. Neonates are particularly sensitive to unanticipated external stimuli, and repetitive exposure to painful medical procedures can have an effect on the development of the central nervous system [2]. Cohort studies demonstrated that exposure to painful procedures can reduce the volume of white and gray matters in preterm infants [3] and that this adverse effect lasts until adolescence [4,5,6]. Repetitive exposure to painful procedures in neonates can also have an impact on behavioral abnormalities [7,8], as indicated by reduced cognitive score [7,8], as well as internalizing behaviors [9], and these adverse effects last until young adulthood [10,11,12,13,14]. Collectively, these studies suggest that neonates are put at significant risk by being subjected to repetitive painful procedures. Although the mechanism of how pain causes damage to the central nervous system in neonates remains unknown, a study in 1-week-old rats suggested that exposure to repetitive pain can alter the production of proteins involved in the processes of development by inducing apoptosis of crucial neurons [15].

Heel lance is one of the most commonly performed procedures associated with pain [16,17]. Multiple interventions such as pacifiers, facilitated tucking, holding, and music are used alone or in combination to reduce the impact of the pain associated with heel lance in neonates [18,19,20,21,22,23,24,25].

Most studies that examined pain relief strategies for heel lance used either the premature infant pain profile (PIPP) or premature infant pain profile—revised (PIPP-R) as a measure of pain. PIPP consists of two physiological measures (heart rate and oxygen saturation) and three behavioral measures (brow bulge, eye squeeze, and nasolabial furrow), and the score is corrected for gestational age (GA) and behavioral state [26,27]. PIPP-R further revised the weights to adjust for GA and behavioral state [28,29]. Both PIPP and PIPP-R were shown to be reliable and valid [26,28,29,30]. However, a study demonstrated that PIPP has poor sensitivity in detecting responses to low-intensity noxious stimuli [31]. Thus, an objective and quantitative measure of pain is needed to properly assess the pain associated with heel lance in neonates to examine the effectiveness of various pain relief strategies.

Event-related potential (ERP) evoked by nociceptive stimulation has recently been attracting attention as an index of pain assessment in neonates [32,33,34,35,36]. Noxious stimuli activate peripheral nociceptors and pain-sensing neurons in the skin, and myelinated Aδ and unmyelinated C fibers are the main peripheral nociceptors [37,38,39]. Aδ fibers respond to rapid, pricking, and localized pains, while C fibers respond to diffuse burning or aching sensations [37,40,41,42,43]. Noxious stimuli detected by peripheral nociceptors are transmitted to the cerebral cortex via the spinal cord, brainstem, and thalamus [44,45]. ERP detects changes in electrical activity generated by neurons in the brain.

In adults, it has become possible to selectively stimulate Aδ and C fibers with intraepidermal electrical stimulation (IES) and laser stimulation and to record evoked potentials [46]. Furthermore, it has been reported that the amplitude of ERP evoked by IES and laser stimulation decreased after analgesic administration in adults [47,48]. Thus, ERP evoked by nociceptive stimuli can be used as a tool for assessing the efficacy of pain management. In regard to neonates, ERP evoked by heel lance was initially reported in 2010 [49], and it was considered to be an objective measure of pain in neonates being unable to express pain in language. However, to our knowledge, there has been no systematic review of the results of ERP evoked by heel lance in neonates. Only a scoping review [50] and unsystematic review articles [32,33,34,36] have been reported previously. In the scoping review, the absence of sample size calculation and problems with statistical analysis have been pointed out [50]. A systematic review in this area will help to evaluate the existing evidence and provide a platform to identify future research needed to extend our knowledge.

In this study, to clarify the possibility of ERP evoked by heel lance in neonates as an index of pain assessment, knowledge acquired by and problems of the methods used in studies on ERP evoked by heel lance in neonates were systematically reviewed, including knowledge about Aδ and C fibers responding to noxious stimuli and Aβ fibers responding to non-noxious stimuli. A better understanding of the mechanisms underlying pain responses in the central nervous system of neonates will lead to the development of an appropriate measure of pain in neonates.

## 2. Materials and Methods

### 2.1. Statement on Review

The present review was performed in accordance with the Preferred Reporting Items for Systematic Reviews and Meta-Analyses (PRISMA) guidelines [51,52].

### 2.2. Search Strategy

PubMed, CINAHL, PsycINFO, Scopus, and CENTRAL were searched until January 2021 with no start date. Keywords for the search were (“heel lance” OR “heel stick” OR “heel prick” OR “heel puncture” OR “pain”) AND (“EEG” OR “ERP” OR “electroencephalogram” OR “event related potential”) AND (“neonate” OR “infant” OR “newborn”). Studies that met the following criteria were included: (1) studies that evaluated whether ERP was appropriate as an objective measure of nociception induced by heel lance in neonates, or (2) randomized controlled trials (RCTs) or non-RCTs that used ERP to determine the efficacy of pain relief strategies to reduce pain associated with heel lance in neonates. Studies were also included if they evaluated ERP itself or in comparison to other measures. Both preterm and full-term infants were included in our definition of neonates.

### 2.3. Selection Criteria

The following inclusion criteria were used based on the PICOS model. (P) population: preterm and full-term infants admitted to NICU; (I) intervention: whether ERP is appropriate for pain evaluation; (C) comparison: with another pain assessment tool (PIPP, behavioral state, near-infrared spectroscopy (NIRS), and/or electromyography (EMG)); (O) outcomes: ERP responses; (S) study design: RCT and non-RCT. Studies were excluded from the review if they (1) assessed pain that is not associated with heel lance, (2) used pain assessment tools other than ERP, or (3) were written in a language other than English.

### 2.4. Study Selection Process

#### 2.4.1. Primary Screening

Two independent investigators (Y.S. and I.S.) screened the literature based on the title and abstract, and duplicates were removed from a list of studies. When the two investigators disagreed on the inclusion of a study, they discussed it. When it could not be resolved by discussion, a third investigator (H.U.) made the decision on whether to include or exclude the study.

#### 2.4.2. Secondary Screening

Based on the primary screening, two investigators (Y.S. and H.U.) independently identified studies that met the inclusion criteria. When the two investigators disagreed on the inclusion of a study, they discussed it. When it could not be resolved by discussion, a third investigator (I.S.) made the decision to include or exclude the study.

### 2.5. Assessment of the Methodological Quality of the Studies

The quality of the studies was assessed independently by two investigators (Y.S. and H.U.). Some of the studies were discussed between the two investigators, and a third investigator (M.O.) made the decision to include or exclude the study when needed. In four studies investigating whether ERP can be used for pain assessment in neonates by recording brain activities in response to heel lance required for particular treatments and to harmless tactile stimuli, the study protocol was regarded as observational by the authors [53,54,55,56]. For the purpose of our review, these pain-inducing procedures were considered as interventions, and we categorized these four studies, in addition to 10 other studies with no study design information, as non-RCTs (comparative and non-comparative study design).

The quality of non-RCTs was evaluated using the methodological index for non-randomized studies (MINORS) [57]. MINORS assesses the methodological quality of non-RCTs and is applicable to both comparative and non-comparative studies. It consists of the following 12 items, of which the first eight are applicable to non-comparative studies while all are applicable to comparative studies: (1) a clearly stated aim; (2) inclusion of consecutive patients; (3) prospective collection of data; (4) endpoints appropriate to the aim of the study; (5) unbiased assessment of the study endpoint; (6) follow-up period appropriate to the aim of the study; (7) loss to follow up less than 5%; (8) prospective calculation of the study size; (9) an adequate control group; (10) contemporary groups; (11) baseline equivalence of groups; and (12) adequate statistical analyses. Each item is scored between 0 and 2 (0: not reported; 1: reported but inadequate; or 2: reported and adequate), and the global ideal scores are 24 and 16 for comparative and non-comparative studies, respectively. It is a reliable and validated system [57] and has been used in various reviews [58].

The methodological quality of RCTs was evaluated using the modified Jadad quality scale [59,60,61]. Its evaluation is based on whether a study includes the description of the following six items: (1) randomization; (2) double-blinding; (3) withdrawals and dropouts; (4) inclusion and exclusion criteria; (5) method used to assess adverse effects; and (6) statistical analyses used. Each item is scored either 0 (no) or 1 (yes), with a positive answer getting a point. Furthermore, an additional point is given for items 1 and 2 if the methods of randomization and blinding are appropriate, and a point is deducted if the methods are inappropriate. Studies with overall scores of 0–3 and 4–8 are considered to be of low and high quality, respectively [60,62]. The modified Jadad quality scale has been used in a number of systematic reviews [60], including a systematic review on the topic of pain assessment scale in children [63].

### 2.6. Data Extraction

Data were extracted from studies independently by two investigators (Y.S. and H.U.). The two investigators discussed differences in data analysis when needed, and a third investigator (K.Y.) was involved in the decision-making process in some cases. The following information was extracted from non-RCTs: age at birth of participants; age at time of the study of participants; number of participants; stimulation; outcomes, and results. Similarly, the following information was extracted from RCTs: age at birth of participants; age at time of the study of participants; number of participants; stimulation; intervention; outcomes, and results.

## 3. Results

### 3.1. Study Selection

A total of 863 studies were extracted from the databases, and 19 were included in the review (Figure 1). Five of those studies were RCTs [64,65,66,67,68] and the remaining 14 were non-RCTs. Among the non-RCTs, five were comparative studies [53,55,69,70,71] and the remaining nine were non-comparative studies [31,49,54,56,72,73,74,75,76].

### 3.2. Study Characteristics

Table 1 lists the MINORS scores for 14 non-RCTs and Table 2 lists the modified Jadad scale score for 5 RCTs. Three RCTs were considered to have high methodological quality. Table 3 and Table 4 summarize the studies included in the review. The GA of the study subjects ranged from 23 to 42 weeks at birth and 28 to 47 weeks at the time of the study. A total of 854 neonates admitted to an NICU were included in the analysis. One of the studies included only preterm infants [53], 11 included both preterm and full-term infants [49,55,56,64,68,69,70,72,74,75,76], and 6 included only full-term infants [31,54,65,66,67,71]. One study did not indicate the GA of the study subjects [73].

Four studies examined the response to heel lance alone [53,64,70,75] and nine examined the response to heel lance and tactile stimulus (contact of a heel lance device against the skin [49,66,71,74,76], rubber bung [69], tendon hammer [72,73,74], air puff [54], or cold puff [54]). Other studies examined the response to the following stimuli in addition to heel lance: tactile stimulation (contact of a heel lance device against the skin) and auditory stimulation [56], experimental noxious stimulation (pinprick) [31,65], experimental noxious stimulation (pinprick) and blood sampling [67], experimental noxious stimulation (pinprick), tactile stimulation (tendon hammer), visual stimulation, and auditory stimulation [55], and tactile stimulation (contact of a heel lance device against the skin) and retinopathy of prematurity screening examination [68].

The outcome measures included in the studies were as follows: ERP alone [49,64,69,70,71,72]; ERP and either PIPP or PIPP-R [67,68,76]; ERP, PIPP, and EMG [31,66]; ERP, PIPP, and heart rate [55]; ERP, PIPP, facial expression, heart rate, and oxygenation [56]; ERP and EMG [65]; ERP and NIRS [74]; ERP, EMG, NIRS, ECG, behavioral response, and autonomic responses (heart rate, oxygen saturation, respiratory rate, and cardiovascular activity) [73]; ERP, facial response, and heart rate [64]; ERP and crying [54]; ERP, PIPP, salivary cortisol, and heart rate variability [75], and ERP and behavioral indicators of infant pain (BIIP) [53].

An ERP analysis was conducted at electrode Cz (international 10–20 system) in nine studies [31,55,56,65,66,71,74,75,76], at electrode Cpz in two studies [72,73], at electrodes Cz and Cpz in four studies [49,64,69,70], and at electrodes C3, C4, F3, and F4 in one study [54]. Two studies did not indicate the electrode location [67,68].

### 3.3. ERP Evoked by Heel Lance

A total of 17 studies demonstrated that heel lance in neonates evoked a specific ERP waveform that consists of both negative and positive peaks [31,49,54,55,56,64,65,66,68,69,70,71,72,73,74,75,76]. One study demonstrated an elevated frequency after heel lance [67], while another did not demonstrate any significant change in frequency [53].

ERP response evoked by heel lance was correlated with PIPP score [75] and EMG response [31] and did coincide with NIRS [73,74]. It did not coincide with crying [54]. When low-intensity pinprick noxious stimulation was applied to infants, ERP response was significantly different from background data, but no difference was noted in PIPP score [31].

### 3.4. Comparison of ERP Evoked by Heel Lance and Tactile Stimulation

Six studies identified distinct negative (N2) and positive (P2) waves after tactile stimulation (between 100 and 400 ms) [56,69,71,72,73,74]. Ten studies demonstrated that heel lance produced negative (N3) and positive (P3) waves (late component) (between 300 and 700 ms) following N2P2 waves (early component) [49,55,56,69,70,71,72,73,74,75]. Three of the studies provided details of latency, with 420 and 560 ms for negative and positive waves, respectively [49], 383 and 554 ms for negative and positive waves, respectively [74], and 403 and 538 ms for negative and positive waves, respectively [71]. Seven studies reported ERP components only in a period of 400–755 ms after stimulation with heel lance [31,54,64,65,66,68,76].

### 3.5. Association between GA and ERP Evoked by Heel Lance

Four studies reported an association between GA and ERP evoked by heel lance [69,70,72,76]. Two of the studies demonstrated that ERP response was less likely to be observed in preterm infants compared with full-term infants (33% (10/30) at GA 28–36 weeks, and 63% (19/30) at GA 37–45 weeks [72]; 12.5% (1/8) at GA 28–32 weeks, and 82% (27/33) in GA 33.9–42 weeks [76]). The occurrence of ERP response increased with GA, and the critical period was GA 35–36 weeks [72]. The remaining two studies did not demonstrate a significant association between GA and ERP; one study identified a distinct ERP response in the youngest neonates of the study cohort [70], while the other demonstrated that the range of ERP was greater in preterm infants compared with full-term infants [69].

### 3.6. ERP as an Indicator of the Effect of Pain Relief Strategies against Heel Lance in Neonates

Five RCTs used ERP to determine the effect of pain relief strategies against heel lance in neonates. These strategies included sucrose [66], glucose [67], C-tactile (CT) optimal touch in comparison to CT non-optimal touch [65], holding by a parent either skin-to-skin or with clothing [64], and morphine [68]. CT optimal touch and skin-to-skin holding by a parent reduced the ERP evoked by heel lance [64,65]. Sucrose reduced PIPP but not ERP evoked by heel lance [66], and morphine was not effective in reducing PIPP-R or ERP [68]. Norman et al. examined the response to heel lance (skin breaking), heel prick (non-skin-breaking pinprick), and blood collection and administered either glucose or water to subjects only in the heel prick experiment [67]. They demonstrated that the administration of glucose lowered PIPP but not ERP in response to heel prick.

## 4. Discussion

In this study, to clarify the possibility of ERP evoked by heel lance in neonates as an index of pain assessment, knowledge acquired by and problems of the methods used in studies on ERP evoked by heel lance in neonates were systematically reviewed, including knowledge about Aδ and C fibers responding to noxious stimuli and Aβ fibers responding to non-noxious stimuli. Previous reviews were limited to a scoping review, pointing out problems with the sample size and statistical analysis as study methodology [50], and unsystematic mini-reviews [32,33,34,36]. To our knowledge, no review has systematically organized the study results of ERP evoked by heel lance in neonates. The present review clarified problems of the results and methods of studies on ERP evoked by heel lance in neonates, which may lead to the development of a pain assessment index for heel lance pain in neonates.

### 4.1. Results of ERP Evoked by Heel Lance in Term Infants

#### 4.1.1. Characteristics of ERP Evoked by Heel Lance

In term infants at 37 weeks or more of GA at the time of participation in the study, noxious stimulation with heel lance induces two specific negative and positive ERP waves. In a study in which ERP evoked by heel lance was compared with that evoked by non-noxious tactile stimulation, both heel lance and non-noxious tactile stimulation induced negative (N2) and positive (P2) waves at around 100–400 ms (early components). In addition to these, heel lance induced negative and positive waves at around 300–700 ms (late components) [49,55,56,69,70,71,72,73,74,75]. Regarding the late ERP waves evoked by heel lance, Verriotis et al. [74] described that N3P3 is a heel lance-specific ERP following the early components (N2P2), and later studies also reported N3P3 in ERP evoked by heel lance [55,56,70,75]. When heel lance was applied to the same neonate at two different times, the early (N2P2) and late (N3P3) ERP components were consistently observed at both times [49]. On the other hand, seven studies reported that heel lance evoked ERP waves only around 400–755 ms [31,54,64,65,66,68,76]. As the latency of these ERP waves was close to that of N3P3 in other studies, they are likely to be the late components (N3P3). Therefore, noxious heel lance and non-noxious tactile stimulation may be distinguished by the late components (N3P3) in neonates.

#### 4.1.2. Question about Latency of ERP Evoked by Heel Lance in Neonates

Although ERP waves have been demonstrated to be evoked by heel lance in a relatively larger number of previous studies, a question remains with regard to the latency of N3P3. In a study in which lance stimulation was applied to adults using the same lance device as that used for heel lance in neonates, the latency of lance stimulation-evoked ERPs was around 100–130 ms for N2 and around 250 ms for P2 (N2: 102 ms, P2: 249.5 ms [71]; N2: 130 ± 40 ms, P2: 258 ± 61 ms [77]). This latency is extensively short in comparison to that of ERPs evoked by IES, which selectively stimulates the Aδ (N2: 199–232 ms, P2: 302–377 ms) [78,79,80,81,82,83,84] and C fibers (P2: 1006–1578 ms) [82,85]. The latency of lance stimulation-evoked ERPs seems to be rather close to that evoked by non-noxious transcutaneous electrical stimulation (ES), which mainly stimulates Aβ fibers (N2: 134–147 ms, P2: 235–293 ms) [78,79,80]. IES and laser stimulation generate electric currents and selectively stimulate the free nerve ending of Aδ fibers present in the epidermis [80]. The electric current of ES reaches the deeper dermis, where the Aβ receptors are present [80]. Thus, it is possible that heel lance stimulates the Aβ receptors. Specifically, the blade of a lance device used for heel lance reaches a depth of 1 mm from the skin surface, crossing the 0.2-mm epidermal layer where Aδ and C fibers reside [78,86], and enters into the dermis (Aβ receptors) [80]. In addition, the lance device is pressed onto the skin surface during lance, which may activate Aβ fibers (tactile pressure or vibration). Accordingly, it cannot be ruled out that lance stimulation activates the Aβ fibers, causing the latency of ERP waves to be closer to that evoked by Aβ-fiber stimulation.

The latency of N3P3 evoked by heel lance in neonates was reported to be 420 ms [49], 383 ms [74], and 403 ms [71] for N3 and 560 ms [49], 554 ms [74], and 538 ms [71] for P3, being longer than that evoked by Aδ fiber stimulation in adults. However, due to the lower amount of myelination and immature electric current dynamics, neurotransmission of noxious stimuli is slower in neonates than adults [87,88], and there are currently no studies that have specifically examined the latency of ERP evoked by Aδ or C fiber stimulation in neonates. Thus, the exact latency evoked by Aδ or C fiber stimulation or heel lance in neonates is unclear.

Therefore, we propose the possibility that heel lance-evoked ERPs observed in previous studies reflect the activation of Aβ fibers caused by the blade of the lance device reaching the dermis and/or pressure of the device during lance procedure. Further studies are needed to clarify the latency of ERP evoked by Aδ and C fiber stimulations in neonates.

### 4.2. Comparison between ERP and PIPP

PIPP has been used to assess heel lance pain in neonates in combination with facial expression and physiological index in many previous studies. The PIPP provides a heel lance pain score [55,56,66,67,68], and the PIPP score and the amplitude of ERP (N3P3) evoked by heel lance were positively correlated [75]. However, it has been pointed out that the PIPP may not be sensitive enough to detect low-intensity stimuli compared to ERP [31]. When an experimental pinprick stimulation (the force of 32, 64, and 128 mN) was used, the PIPP score did not significantly differ from the pre-experimental period. On the other hand, ERP was evoked by all types of experimental pinprick stimulation, and it was significantly different from the pre-experimental period. This result may indicate that ERP has a higher detection sensitivity for low-intensity stimuli than PIPP. However, the latency of ERP evoked by heel lance in neonates questions the activation of Aδ and/or C fibers. It remains unclear at this time whether ERP is more sensitive than PIPP.

### 4.3. ERP-Based Evaluation of Intervention Effect

In studies using ERP amplitude as an index of pain assessment, CT optimal touch [65] and holding by a parent, skin-to-skin [64], were found to significantly reduce the N3P3 amplitude of ERP evoked by heel lance in neonates. N3P3 is a nociceptive stimulation-specific ERP because it is evoked by heel lance but not by tactile stimulation. Therefore, the decrease in the N3P3 amplitude could indicate that CT optimal touch and skin-to-skin holding by a parent are useful methods to manage heel lance pain in neonates. On the other hand, sucrose [66] and morphine [68] do not decrease the N3P3 amplitude; thus, they were not regarded as a method to relieve pain in neonates.

However, there is a possibility that ERP evoked by heel lance in neonates reflects the activation of Aβ fiber. Thus, assessing heel lance pain in neonates using ERP may not be appropriate. It is important to perform a study clarifying the ERP evoked by Aδ and C fiber stimulations in neonates.

### 4.4. ERP Evoked by Heel Lance in Preterm Infants

A low detection rate of ERP evoked by heel lance in preterm infants can be problematic. In two previous studies, the occurrence of ERP was significantly lower in preterm infants than full-term infants [72,76] and was found to increase with progression of GA [72]. In preterm infants, neuronal bursts can occur in brain circuitry frequently [72,89]. It has been suggested that a transition from non-specific neuronal bursts to specific evoked potentials occurs at 35–37 weeks of GA to achieve a discrimination between touch and nociception [72], indicating that ERP evoked by nociceptive stimuli can be detected in neonates at around 35 weeks GA [72].

On the other hand, one study reported no significant association of the occurrence of ERP with GA [70]. Specifically, the occurrence of ERP at electrodes showing the maximum response (Cz, C4, C3, CP4, and CP3) was not significantly associated with GA, but a finding of the maximum response at the vertex electrode (Cz) was consistent with two previous reports [72,76]. The response was also lower in preterm than term infants [70]. It is possible that activation of multiple brain regions by heel lance caused the large response at midline electrodes [70,90,91]. In preterm infants, a low occurrence of ERP at the vertex electrode has been suggested to be caused by an immaturity of brain, but there are only a few reports that investigated ERPs in preterm infants, necessitating further studies [70].

It has also been reported that ERP evoked by heel lance was larger in preterm infants compared to term infants. One study demonstrated that the amplitude of ERPs evoked by heel lance was greater in preterm infants hospitalized in the NICU for at least 40 days than healthy term infants [69]. On the other hand, the response to light touch was found to be smaller in preterm infants than term infants [92]. There are various interpretations of the amplitude ERP evoked by heel lance. Some studies have proposed that the ERP amplitude reflects the intensity of pain perception [93,94,95,96]. Preterm infants undergo painful treatments frequently during the NICU stay and thus may be more responsive to noxious heel lance and less responsive to non-invasive light touch [92]. In other words, abnormal neurotransmission develops as a consequence of frequent painful treatments in preterm infants. On the other hand, it is possible that ERP amplitude does not reflect the intensity of pain perception but rather captures the magnitude of attention to a stimulus [97,98]. Nevertheless, ERP studies for preterm infants are at the developmental stage, leaving many unclear points. To advance ERP research in preterm infants, it is necessary to clarify activity of pain-sensing neurons and their association with ERP evoked by heel lance firstly in term infants.

### 4.5. Problems of ERP Study Methods in Neonates

A review of pain assessment using neurophysiological measurements (ERP, NIRS, and functional magnetic resonance imaging (fMRI)) for invasive procedures such as heel lance, intramuscular injections, and noxious pinprick stimulation in neonates highlighted some concerns about the sample sizes of ERP studies [50]. Specifically, the authors demonstrated that only two of eight studies reported sample size calculations, suggesting the need for proper sample size calculations [50]. Similarly, we also demonstrated that only 3 of 13 studies reported sample size calculations. These findings highlight the need for appropriate sample size calculations.

In general, studies on ERP report an average from multiple stimulation trials, with *n* = 10 to 12 for intraepidermal electrical stimulation and transcutaneous stimulations [78,79,82,84], to constitute a grand average of the group. On the other hand, the grand average of a single stimulation is calculated for ERP evoked by heel lance in neonates. It is ethically challenging to expose neonates to multiple experimental heel lances, and all of the studies included in the review examined the response to heel lance procedures that were clinically required. Thus, reproducibility of the measurements is a concern in this population [99]. Future studies should address the limitations associated with latency, sample size, and the use of arithmetic means when assessing the level of pain using ERP as an indicator. There is an increasing interest in the brain network associated with pain stimulation in neonates. For example, a study used fMRI to examine the brain activity in response to pinprick in neonates and demonstrated that the areas activated by pain were similar to those in adults [100]. Since ERP can be limited to a few electrodes, it could be beneficial to examine the entire brain network in future studies.

Therefore, problems also remain in the method of studies on ERP evoked by heel lance in neonates, and it is necessary to perform a study with a sufficient sample size and multiple heel lance stimulations.

There are several limitations to our review. First, since only three studies reported sample size calculations, most of the results were not supported by a validated sample size. Exclusion of studies that were not written in English may also have an impact on the overall results of the review.

## 5. Conclusions

ERP evoked by heel lance in neonates consists of N2P2 and N3P3 waves, and N3P3 appears to be induced specifically by heel lance. However, it is possible that N3P3 reflects the activation of Aβ fibers responding to non-noxious stimuli. Furthermore, there are methodological problems such as non-calculated sample size and small number of stimulation trials to obtain an averaged ERP. To establish ERP evoked by heel lance as an index of pain assessment in neonates, it is necessary to clarify ERP evoked by Aδ- and C-fiber stimulations accompanied by heel lance in term infants. A better understanding of this aspect may lead to the development of a method reliving heel lance pain and of a pain assessment index for preterm infants.

## Figures and Tables

**Figure 1 children-08-00058-f001:**
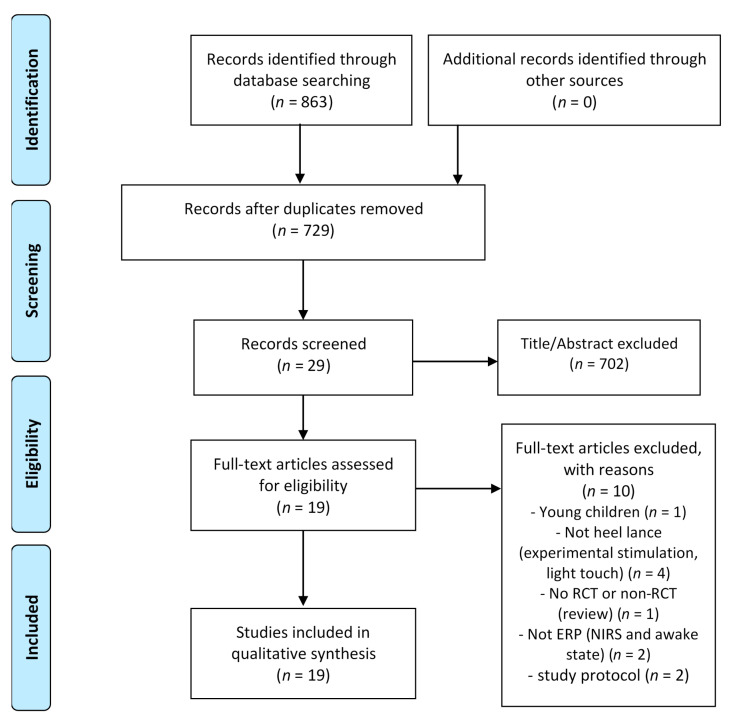
Flow chart of included article.

**Table 1 children-08-00058-t001:** Assessment of the methodological quality of the non-randomized controlled trial (non-RCT) (methodological index for non-randomized studies (MINORS) score).

Author, Year	Study Design	A Clearly Stated Aim	Inclusion of Consecutive Patients	Prospective Collection of Data	Endpoints Appropriate to the Aim of the Study	Unbiased Assessment of the Study Endpoint	Follow-up Period Appropriate to the Aim of the Study	Loss to Follow Up Less than 5%	Prospective Calculation of the Study Size	An Adequate Control Group	Contemporary Group	Baseline Equivalence of Groups	Adequate Statistical Analyses	Total
Slater, 2010 [69]	Comparative	2	1	2	2	0	0	0	0	2	2	2	2	15
Maimon, 2013 [53]	Comparative	2	2	2	1	1	0	0	2	1	1	1	2	15
Fabrizi, 2016 [71]	Comparative	2	2	2	2	0	0	0	0	1	0	1	2	12
Hartley, 2017 [55]	Comparative	2	1	2	1	0	0	1	2	1	0	1	2	13
Verriotis, 2018 [70]	Comparative	2	2	2	2	0	0	0	0	1	2	1	2	14
Slater, 2010 [49]	Non-comparative	2	1	2	1	0	0	0	0	–	–	–	–	6
Fabrizi, 2011 [72]	Non-comparative	2	2	2	1	0	0	0	0	–	–	–	–	7
Worley, 2012 [73]	Non-comparative	1	0	1	1	0	0	0	0	–	–	–	–	3
Hartley, 2015 [31]	Non-comparative	2	2	2	1	1	0	0	0	–	–	–	–	8
Verriotis, 2016 [74]	Non-comparative	2	2	2	2	0	0	0	0	–	–	–	–	8
Maitre, 2017 [54]	Non-comparative	2	2	2	1	0	0	0	2	–	–	–	–	9
Jones, 2017 [75]	Non-comparative	1	1	2	1	1	0	0	0	–	–	–	–	6
Jones, 2018 [56]	Non-comparative	1	1	2	1	0	0	0	0	–	–	–	–	5
Green, 2019 [76]	Non-comparative	2	2	2	1	1	0	1	0	–	–	–	–	9

RCT: randomized controlled trial.

**Table 2 children-08-00058-t002:** Assessment of the methodological quality of the RCT (Jadad score).

Author, Year	Randomization	Blinding	Withdrawals and Dropouts	Inclusion/ Exclusion Criteria	Adverse Effects	Statistical Analysis	Total
Norman, 2008 [67]	1	1	0	1	0	1	4
Slater, 2010 [66]	2	2	1	1	1	1	8
Gursul, 2018 [65]	1	0	0	0	0	1	2
Hartley, 2018 [68]	2	1	1	1	1	1	7
Jones, 2020 [64]	0	0	0	1	0	1	2

RCT: randomized controlled trial.

**Table 3 children-08-00058-t003:** Summary of included studies (non-RCT).

Author, Year	Study Design	Participants			Stimulation	Outcomes Measured	Results	* Quality Score
		Age at Birth	Age at Time of the Study	Number				
Slater, 2010 [49]	Non- comparative	35–39 weeks PMA	2–13 days	10	Heel lance Non-noxious control (contact of a heel lance device against the skin without the skin being touched by the blade)	ERP	ERP evoked by heel lance was different from that by non-noxious control stimulation.	6/16
Slater, 2010 [69]	Comparative	Term infants: 37–40 weeks PMA, Preterm infants: 24–32 weeks PMA	Term infants: 37–41 weeks PMA, Preterm infants: 37–41 weeks PMA	15 (term infants: *n* = 8, preterm infants: *n* = 7)	Heel lance Lightly tapping a rubber bung	ERP	ERP evoked by heel lance was significantly larger in preterm infants than term infants.	15/24
Fabrizi, 2011 [72]	Non- comparative	24–42 weeks GA	28–46 weeks GA	46	Heel lance Lightly tapping a tendon hammer	ERP	In full-term infants, ERP evoked by heel lance (300–700 ms) was different from that by tactile stimulation (50–300 ms). The percentage of occurrence of ERP by both heel lance and tactile stimulation was significantly smaller in the preterm infants than the full-term infants.	7/16
Worley, 2012 [73]	Non- comparative	Infants	No data	6	Heel lance Lightly tapping a tendon hammer	ERP EMG NIRS ECG Behavioral responses Autonomic responses (heart rate, oxygen saturation, respiratory rate, and cardiovascular activity)	ERP evoked by heel lance consisted of an early component followed by a late component. Tapping stimuli evoked only the early component. ERP evoked by heel lance was coupled with NIRS response. Only heel lance elicited a larger flexion withdrawal reflex and behavioral responses, increased heart rate, and decreased oxygen saturation.	3/16
Maimon, 2013 [53]	Comparative	Group 1: 27–29 weeks GA, Group 2: 27–29 weeks GA, Group 3: 32–33 weeks GA	Group 1: 30 weeks GA; <10 days PNA, Group 2: 33 weeks GA, Group 3: 34.1 weeks GA	Group 1: *n* = 24, Group 2: *n* = 22, Group 3: *n* = 25	Heel lance	ERP (evoked power) Behavioral indicators of infant pain	No significant difference in brain activity was found between pre- and post-heel lance.	15/24
Hartley, 2015 [31]	Non- comparative	Term infants	37–42 weeks GA; <10 days PNA	30	Heel lance Noxious stimulation (pinprick: 32 mN, 64 mN, 128 mN)	ERP PIPP EMG	ERP evoked by heel lance (400–700 ms) was greater than that by non-noxious control stimulation. The magnitude of ERP was significantly correlated with the magnitude of EMG. The ERP and EMG magnitudes increased with stimulus intensity of pinprick.	8/16
Verriotis, 2016 [74]	Non- comparative	36.3–42.0 weeks GA	36.6–43.3 weeks GA; 0–16 days PNA	30	Heel lanceControl stimulation (contact of a heel lance device against the skin without the skin being touched by the blade) Tactile stimulation (lightly tapping a tendon hammer)	ERP NIRS	ERP evoked by heel lance consisted of N2P2 waves (139 and 202 ms) followed by N3P3 waves (385 and 554 ms). Tactile stimulation induced only N2P2 waves (147 and 248 ms). Electrophysiological (ERP) and hemodynamic responses (NIRS) by heel lance coincided and were positively correlated.	8/16
Fabrizi, 2016 [71]	Comparative	Infants: 37–42 weeks GA, Adults: –	Infants: 0–19 days PNA (5.8 ± 4.3), Adults: 23–48 years (29.7 ± 6.0	Infants: 18, Adults: 21	Infants: Heel lance, Control stimulation (contact of a heel lance device against the skin without the skin being touched by the blade) Adults: Noxious stimulation (a sterile lancet to prick the fifth finger), Control stimulation (contact of a heel lance device against the skin)	ERP	Heel lance (or noxious stimulation) and control stimulation evoked N2P2 waves in infants (140 and 225.5 ms for heel lance and 151.5 and 227 ms for control stimulation) and in adults (102 and 249.5 ms for noxious stimulation and 93.5 and 180.5 ms for control stimulation). Heel lance also evoked a N3P3 waves (403 and 538 ms) in infants but not in adults.	14/24
Maitre, 2017 [54]	Non- comparative	37–42 weeks GA	2–3 days PNA	54	Heel lance Light touch (air puff) Cold puff	ERP Crying	ERP evoked by heel lance consisted of a late component. ERP evoked by heel lance was not associated with either the presence or amplitude of cries.	9/16
Jones, 2017 [75]	Non- comparative	36–42 weeks GA	No data	56	Heel lance	ERP Salivary cortisol Heart rate variability PIPP	ERP evoked by heel lance consisted of N3P3 waves. ERP amplitude was significantly correlated with PIPP.	6/16
Hartley, 2017 [55]	Non- comparative	31.9–41.4 weeks GA	35.1–43.6 weeks GA	72	Heel lance Experimental noxious stimulation (128 mN, pinprick; MRC systems) Experimental tactile stimulation (modified tendon hammer) Visual stimulation Auditory stimulation	ERP Heart rate PIPP (facial expression)	ERP evoked by heel lance (400–700 ms) was different from that evoked by non-noxious tactile stimulation.	13/24
Verriotis, 2018 [70]	Non- comparative	29–42 weeks GA	29–43 weeks GA; no older than 2 weeks PNA	81	Heel lance	ERP	ERP evoked by heel lance consisted of N2P2 waves followed by N3P3 waves (400–700 ms). Females were more likely to exhibit a widespread ERP than males.	14/24
Jones, 2018 [56]	Non- comparative	23–42 weeks GA	29–47 weeks GA (0–96 days)	112	Heel lance Non-noxious sham (contact of a heel lance device against the skin without the skin being touched by the blade Auditory controls	ERP Facial expression Heart rate Oxygenation PIPP	ERP evoked by heel lance consisted of N2P2 waves followed by N3P3 waves. ERP evoked by non-noxious sham and auditory control consisted of N2P2 waves only.	5/16
Green, 2019 [76]	Non- comparative	23–42 weeks GA	28–42 weeks GA	49	Heel lance Control lance (contact of a heel lance device against the skin without the skin being touched by the blade)	PIPP-R (facial expression) ERP	ERP evoked by heel lance was different from that by non-noxious control stimulation. The occurrence of ERP evoked by heel lance increased with GA.	9/16

* Assessment of the methodological quality by MINORS score. RCT: randomized controlled trial; PMA: postmenstrual age; ERP: event-related potential; GA: gestational age; EMG: electromyography; NIRS: near-infrared spectroscopy; ECG: electrocardiography; N: newton; PNA: postnatal age; PIPP: Premature Infant Pain Profile; PIPP-R: Premature Infant Pain Profile—Revised.

**Table 4 children-08-00058-t004:** Summary of included studies (RCT).

Author, Year	Participants			Stimulation	Intervention	Outcomes Measured	Results	* Quality Score
	Age at Birth	Age at Time of the Study	Number					
Norman 2008 [67]	37–42 weeks GA	37–143 h PNA	72	Heel lance Heel prick (non-skin-breaking pin-prick) Venous blood sampling from the dorsum of the hand	(Only heel prick) Glucose Water	ERP (evoked power)PIPP	All noxious stimuli induced a significant increase in higher frequency components (10–30 Hz). (Only heel prick) There was no difference in brain activity between infants who received either glucose or water. The PIPP score was significantly lower in infants who received glucose than those who received water.	4
Slater 2010 [66]	37–43 weeks PMA	>8 days PNA	44	Heel lance Non-noxious control (contact of a heel lance device against the skin)	Sucrose Sterile water	ERP PIPP EMG	ERP evoked by heel lance was significantly greater than that evoked by non-noxious control. ERP and EMG response did not differ significantly between infants who received either sucrose or sterile water. The PIPP score was significantly lower in infants who received sucrose than those who received sterile water.	8
Gursul 2018 [65]	37–42 weeks GA	1–5 days PNA	30	Heel lance Experimental noxious stimulus (128 mN, pinprick; MRC systems)	C-tactile (CT) optimal touch (brush velocity 30 cm/s) CT non-optimal touch (brush velocity 30 cm/s) No-touch control	ERP EMG	CT optimal touch significantly reduced ERP evoked by heel lance, as compared to no-touch control. CT non-optimal touch did not reduce ERP evoked by heel lance.	2
Hartley 2018 [68]	34–42 weeks GA	34.3–36.3 weeks GA 1–20 PNA	31	Heel lance Non-noxious control (contact of a heel lance device against the skin) Retinopathy of prematurity screening examination	Morphine Placebo	ERP PIPP-R	ERP evoked by heel lance was significantly greater than that by non-noxious control. ERP and PIPP-R did not differ between infants who received either morphine or placebo.	7
Jones 2020 [64]	23–41 weeks GA	0–96 PNA	27	Heel lance	While held by a parent in skin-to-skin While held by a parent with clothing Not held at all	ERP Facial response HR	Heel lance evoked noxious ERP components (497 to 755 ms). ERP was significantly lower in infants held skin-to-skin compared to those held with clothes. Facial response score and HR were higher in infants held with clothing than those not held or those held skin-to-skin.	2

* Assessment of the methodological quality by Jadad score. RCT: randomized controlled trial; PMA: postmenstrual age; PNA: postnatal age; ERP: event-related potential; PIPP: Premature Infant Pain Profile; EMG: electromyography; GA: gestational age; N: newton; PIPP-R: Premature Infant Pain Profile—Revised; HR: heart rate.

## Data Availability

Data sharing not applicable.

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
