# Peer review of "Can Event-Related Potentials Evoked by Heel Lance Assess Pain Processing in Neonates? A Systematic Review"

_children, 2021, doi:10.3390/children8020058_

Round 1
Reviewer 1 Report
I found the premise on which this review is based to be interesting, given the lack of effective techniques for assessing pain in early infancy, and the resulting lack in appropriate analgesia (and detriments to neurological development and infant mental heath). PRISMA review criteria were well followed, even if not always explicitly written as such, and the review findings are appropriately described. I am unfamiliar with MINORS and Jadad, but these appear useful, solid, and effective. PRISMA table is well done, and quality reporting tables are clear. For table 3, the summary of study info, I would recommend changing the column heading "outcomes" to "variables measured" or "outcomes measured" for clarity in the English language. Also would be useful to indicate the study's quality rating in a column in table 3 for ease of viewing and access.
Reviewer 2 Report
This manuscript is a systematic review of the evaluations of event-related potentials (ERP) to quantify pain associated with heel lance in neonates.
The manuscript as a whole is coherent, although I have several issues with it, and I believe it should be substantially revised.
The abstract is incomplete as it does not really sums up the content of the results and conclusions. What is the core result of the research?
The introduction is rather short and to the point. I would have appreciated a deeper explanation of the ERP's principle.
Material and Methods
The study design is simple and uses adapted methodology for systematic reviews. The inclusion/exclusion process was limited to 3 authors, but not always in the same order. The qualitative features for incertain decisions could be mentioned.
I have some reserves regarding search terms, which do not include “pain” for example, or the plain text versions of EEG and ERP. These choices are difficult to judge because a particularly broad search can be refined. The search terms are correctly organized.
The rest of the evaluation criteria is clearly stated, including the algorithm.
Results
The results are relatively detailed.
The tables do not mention the methodologies acronym they present.
Discussion
This is my main concern in this manuscript. The discussion does not efficiently discuss the results, and avoids certain topics.
L239: from the tables in the results, only 4 studies mentioned N3P3, therefore your conclusion on N3P3 response (although I understand what the authors are trying to say) should be modulated to consider this. I think a rephrasing of this paragraph is necessary to improve the argumentation here.
L255: This is unfortunate phrasing. Neonates do not lack the process of myelination. Myelination begins well into the second trimester, and continues late after birth, depending on the region.
As it is, this manuscript is still relatively superficial regarding the matter of interest. The results are not discussed in enough details. The tables are very useful and welcome, but the discussion should address discrepancies between studies, and critically analyze the data extracted such as the noted sensibility issues of PIPP compared to ERP, or the different electrodes’ positions and results, and whether they present coherent findings. Similarly, the authors do not mention the existence of other methods for blood collection (doi:10.1002/14651858.CD001452.pub4).
English can be edited in a few spots (see PDF)

Round 2
Reviewer 2 Report
The authors substantially changed the manuscript. The discussion is much more detailed.
I have only few minor corrections.
- Despite the response of the authors, I do not see the MINORS mention in Table 1 title. Table 2 title has been edited.
- L248: It did not coincide WITH crying.
- L298: were limited to a scoping review in pointing out
- L299: mini-reviews.
- L299: To OUR knowledge
- L300 (end): The present review (this review)
- L332: phrasing : "also attracted attention the most" is incorrect.
- L356: is remains unclear
This is only the ones I spotted, but please check the overall English.
